# Novel Insight into the Photophysical Properties and 2D Supramolecular Organization of Poly(3,4-ethylenedioxythiophene)/Permodified Cyclodextrins Polyrotaxanes at the Air–Water Interface

**DOI:** 10.3390/ma16134757

**Published:** 2023-06-30

**Authors:** Alae El Haitami, Ana-Maria Resmerita, Laura Elena Ursu, Mihai Asandulesa, Sophie Cantin, Aurica Farcas

**Affiliations:** 1Laboratory of Physical Chemistry of Polymers and Interfaces, CY Cergy Paris Université, F95000 Cergy, France; alae.el-haitami@cyu.fr (A.E.H.); sophie.cantin-riviere@cyu.fr (S.C.); 2“Petru Poni” Institute of Macromolecular Chemistry, Romanian Academy, Grigore Ghica Voda Alley, 41A, 700487 Iasi, Romania; resmerita.ana@icmpp.ro (A.-M.R.); ursu.laura@icmpp.ro (L.E.U.); asandulesa.mihai@icmpp.ro (M.A.)

**Keywords:** PEDOT, permethylated cyclodextrins, smart optical materials, quantum efficiency, surface morphology, monolayers, optoelectronics

## Abstract

Two poly(3,4-ethylenedioxythiophene) polyrotaxanes (PEDOT∙TMe-βCD and PEDOT∙TMe-γCD) end-capped by pyrene (Py) were synthesized by oxidative polymerization of EDOT encapsulated into TMe-βCD or TMe-γCD cavities with iron (III) chloride (FeCl_3_) in water and chemically characterized. The effect of TMe-βCD or TMe-γCD encapsulation of PEDOT backbones on the molecular weight, thermal stability, and solubility were investigated in depth. UV–vis absorption, fluorescence (F_L_), phosphorescence (P_H_), quantum efficiencies, and lifetimes in water and acetonitrile were also explored, together with their surface morphology and electrical properties. Furthermore, dynamic light scattering was used to study the hydrodynamic diameter (DH) and z-potential (ZP-ζ) of the water soluble fractions of PEDOT∙TMe-βCD and PEDOT∙TMe-γCD. PEDOT∙TMe-βCD and PEDOT∙TMe-γCD exhibited a sharp monodisperse peak with a DH of 55 ± 15 nm and 122 ± 32 nm, respectively. The ZP-ζ value decreased from −31.23 mV for PEDOT∙TMe-βCD to −20.38 mV for PEDOT∙TMe-γCD, indicating that a negatively charged layer covers their surfaces. Surface pressure–area isotherms and Brewster angle microscopy (BAM) studies revealed the capability of the investigated compounds to organize into sizeable and homogeneous 2D supramolecular assemblies at the air–water interface. The control of the 2D monolayer organization through the thermodynamic parameters of PEDOT∙TMe-βCD and PEDOT∙TMe-γCD suggests potential for a wide range of optoelectronic applications.

## 1. Introduction

Among all the materials for organic electronics, conjugated polymers (CPs) have been widely recognized as semiconductor materials due to the possibility of their application in optoelectronic devices [1,2,3]. In particular, poly(3,4-ethylenedioxythiophene) (PEDOT) is an interesting p-type semiconducting material whose thermal, optical, and electronic properties are of great interest for application in various optoelectronic fields [4,5]. Since the solubility of PEDOT compound is a critical element for the application, a lot of synthetic procedures have been developed with a view to improving its solubility and processability [5,6]. With respect to different synthetic methods that are available, new disadvantages appeared, such as limited interchain charge mobility and the rigidity of chain structure [7,8,9,10,11,12]. Currently, the preferred method for the improvement of CPs solubility is encapsulation of their backbones into the macrocyclic cavities via noncovalent interactions resulting in so-called encapsulated CPs [13,14,15]. Previous studies have shown that the threading of macrocyclic molecules onto the conjugated backbones does not disrupt the π-conjugation system but improves the solubility, as well as the morphological characteristics of the resulting encapsulated compounds when compared with their pristine counterpart [14]. More than that, this strategy has been shown to increase the photophysical properties of CPs providing evidence for a diminished tendency toward aggregation of organic semiconductors. The preparation of such encapsulated systems consists in the threading of macrocyclic compounds (hosts) onto monomer/polymer (guests) when pseudopolyrotaxanes (PPs) are obtained. To generate polyrotaxane (PR) architectures, bulky stoppers covalently attached at both ends of the CPs’ backbones are required. As a result of significant interest in manipulating the distance between CPs backbones and preventing crosstalk, a relatively large number of PP and PR architectures containing different host molecules have been reported [16,17,18,19,20,21,22]. These supramolecular materials exhibit a crucial importance for further development of organic electronics and represent a key bottom-up strategy to build and process relatively soft functional materials. It has been shown that this approach can be used for a diminished tendency toward aggregation and to obtain a wide variety of supramolecular architectures by combination of different conjugated guest and macrocyclic host molecules [15,23,24]. Alongside other macrocyclic hosts, several types of permodified cyclodextrins (CDs) derivatives were used for the synthesis of PP and PR architectures [16,17,21,22]. Due to their hydrophobic cavities and surfaces, these host molecules have the ability to thread appropriately sized monomers and oligomers/polymers by cooperation of various noncovalent interactions, such as hydrophobic, electrostatic, or van der Waals. Accordingly, the establishments of functionalized CDs have potential for use in unprecedented encapsulated CPs and the larger hydrophobic permodified CD surfaces can encapsulate neutral guest molecules by hydrophobic interactions similar to those of native CDs. The presence of permodified CDs results in improved solubility in either polar or nonpolar organic solvents, which allows their processing by spin coating [17,21]. Thus, our efforts are being made in this direction and we report here the preparation of PEDOT∙TMe-βCD and PEDOT∙TMe-γCD PR architectures using a slightly modified synthetic procedure consisting in the termination of the growing PEDOT chains with bulky Py groups instead of triphenylmethane [22]. Moreover, taking into consideration that the presence of hydrophobic methyl chains on CD surfaces could greatly affect the supramolecular arrangements of the encapsulated PEDOT∙TMe-βCD and PEDOT∙TMe-γCD compounds, we investigated their ability to organize into larger and homogeneous 2D supramolecular assemblies. The current study shows, for the first time to the best of our knowledge, that these encapsulated PEDOT compounds form stable monolayers at the air–water interface, which has great implications in the field of organic semiconductors for optoelectronics. The chemical structures of the investigated compounds are illustrated in Figure 1.

Therefore, the oxidative polymerization of 3,4-ethylenedioxythiophene (EDOT) in the form of its inclusion complex with TMe-βCD (EDOT∙TMe-βCD) or TMe-γCD (EDOT∙TMe-γCD) as starting monomers in water with FeCl_3_ catalyst enabled the synthesis of unstable PEDOT PPs. To avoid the dethreading of macrocycles, the resulting PPs were converted into PR architectures by the coupling reaction of the PEDOT ends with bulky Py groups. The chemical structures of PEDOT∙TMe-βCD and PEDOT∙TMe-γCD were proven by FT-IR and ^1^H-NMR spectroscopic techniques. Moreover, the molecular weights, thermal stability, photophysics, surface morphology, DH, ZP-ζ, and conductivity were further investigated. Their supramolecular organizations were studied through compression isotherms of Langmuir monolayers formed at the air–water interface and subsequently, their morphologies were analyzed in each stage by BAM analysis. 

## 2. Materials and Methods

### 2.1. Materials

3,4-Ethylenedioxythiophene (EDOT) was purchased from Sigma-Aldrich (St. Louis, MA, USA) and purified before use by vacuum distillation. Py, anhydrous FeCl_3_, and ferrocene (Fc) were purchased from Sigma-Aldrich and used as received. 2′,4′,6′-trihydroxyacetophenone monohydrate (99.5%) purchased from Supelco (St. Louis, MA, USA) and tetrabutylammonium perchlorate (TBAClO_4_) for electrochemical analysis (99.0%) (Fluka, London, UK) were used without further purification. Acetonitrile (ACN) (Fischer, Zürich, Switzerland), methylene chloride (DCM), petroleum ether, toluene, tetrahydrofuran (THF), and all other solvents were purchased from commercial sources (Sigma-Aldrich, Fisher) and used without further purification. 

### 2.2. Characterization

The FT-IR (KBr pellets) spectra were obtained on a Bruker Vertex 70 spectrophotometer. The ^1^H-NMR spectrum was recorded on a Bruker Avance NEO 400 MHz instrument equipped with a 5 mm QNP direct detection probe and z-gradients. The spectrum was recorded in DMSO-d_6_ at ambient temperature. The chemical shifts are reported as δ values (ppm) relative to the solvent residual peaks. The molecular weights of PEDOT∙TMe-βCD and PEDOT∙TMe-γCD compounds were determined by gel permeation chromatography (GPC) with a PL-EMD 950 Evaporative Mass Detector, using as standard polystyrene (Pst) and THF as eluent at flow rate 1 mL⋅ min^−1^. The thermogravimetric analysis was carried out on Mettler Toledo TGA/SDTA 851e equipment (Mettler Toledo, Greifensee, Switzerland) under constant N_2_ flow (20 cm^3^∙min^−1^) and heating rate of 10 °C min^−1^ from 25 to 800 °C. The TGA curves were processed with Mettler Toledo STAR^e^ software (Version 9.10, Giessen, Germany). UV–vis absorption and F_L_ spectra were recorded using a Specord 210 Plus Analytic Jena spectrophotometer and FS5 Edinburg Instruments spectrofluorometer. The time-resolved fluorescence measurements were carried out on an FLS980 spectrofluorometer. The excitation source was a LED with λ_ex_ = 375 nm. Each sample’s lifetime was calculated by analyzing the decays with oxygen. The time-resolved fluorescence and phosphorescence measurements were fitted by using single, double, or triple exponential functions, I(t) = a_1_e ^−t/τ1^ + a_2_e ^–t/τ2^ + a_3_e ^–t/τ3^, where I(t) represents the emission intensity at time t, and a_i_ and τ_i_ are the preexponential factor and the component t of the decay time, respectively. The best fitted parameters were estimated by the minimization of the reduced χ^2^ value and of the residual distribution of the experimental data. Fittings having chi-squared values around 1 and symmetrical distributions of the residual were accepted. The emission quantum yield (Φ) was measured using an FLS980 integrating sphere for a solution at 355 nm as excitation wavelength corresponding to the absorption band. All spectral measurements were performed at room temperature and 10 mm path length quartz cells were utilized for spectroscopic determinations. For transient absorption spectra, all measurements were performed with LP980, Edinburgh Instruments, by using Nd YAG Laser, maxim output 500 mJ, pulse duration 4–6 ns at excitation wavelength 355 nm. LP980 is a fully integrated and sophisticated transient absorption spectrometer that uses the pump–probe technique for measuring transient kinetics, and spectra. In kinetic mode, a single point detector is used to measure the transient kinetics at a single wavelength, and spectra can be built up by automated scanning over a wavelength range and slicing the data. The surface morphology of PEDOT∙TMe-βCD and PEDOT∙TMe-γCD films was highlighted by atomic force microscopy (AFM) using an NTEGRA Spectra (NT-MDT, Russia) instrument with commercially available silicon nitride cantilevers (NSG10, NT-MDT, Russia). A 10 µL measure of the sample was drop-casted on freshly cleaved mica and dried at ambient temperature. Squares of 3 and 3 µm side were scanned in the semi-contact mode in air. The resulting topographical AFM images were analyzed using the software Gwyddion 2.55. Cyclic voltammograms (CVs) were carried out in a three-electrode cell in which bare Pt (1 mm diameter) was used as a working electrode, Pt wire as counter-electrode, and Ag wire as pseudo-reference electrode. The supporting electrolyte was 0.1 M TBAClO_4_ solution in anhydrous ACN. The set-up was controlled by a BIO-LOGIC SP 150 potentiostat/galvanostat using EC-lab software (Version 10.38/August 2014). The pseudo-reference was calibrated with 10^−3^ M Fc solution in ACN for which the apparent redox potential (half-sum of oxidation and reduction potential) was E°_app_ (F_c_/F_c_+) = + 0.434 V vs. Ag wire. Polymer samples were drop-casted onto the working electrode from water and studied in the interval of −0.8 and +1.0 V vs. Ag wire at 20 mV∙s^−1^. The DH and ZP-ζ of PEDOT∙TMe-βCD and PEDOT∙TMe-γCD colloidal dispersions in water were evaluated by a DLS approach at room temperature with a Delsa Nano C-Particle Analyzer (Beckman Coulter, Brea, CA, USA). The particles’ measured electrophoretic mobility was then converted into the ZP-ζ using the von Smoluchowski equation. The results were expressed as the average values of three independent measurements performed for each sample. Dielectric spectroscopy measurements were carried out with the Novocontrol Dielectric Spectrometer (GmbH Germany, Montabauer, Germany), CONCEPT 40, in a wide range of frequencies, 1–10^6^ Hz. The samples were prepared as round pellets with a vacuum force about 10 tons. The pellets with diameter about 13 mm and thickness about 200–400 μm were sandwiched between two gold plated electrodes. Mass Spectrometry: MALDI MS analysis was performed using a RapifleX MALDI TOF TOF MS instrument (Bruker, Bremen, Germany). FlexControl 4.0 and FlexAnalysis 4.0 software (Bruker) were used to control the instrument and process the MS and MS/MS spectra. The sample was prepared by dry kneading 1 mg of the Py-EDOT∙TMe-βCD-Py sample together with 10 mg of 2′,4′,6′-trihydroxyacetophenone monohydrate as a matrix. The MS calibration was performed using poly(ethylene glycol) standards applied to the MALDI MS target. The MS/MS fragmentation experiments were performed in LIFT mode using a Bruker standard fragmentation method. The full isotopic profile of the parent ion was isolated. To elaborate the Langmuir films, all compounds were dissolved in DCM (0.1 g∙L^−1^). These solutions were then sonicated for 5 min in an ultrasonic bath heated to 30 °C and then cooled at room temperature. The subphase consists of ultrapure water produced from a Millipore Simplicity system (Billerica, MA, USA) with a resistivity of 18.2 MΩ∙cm. The Langmuir–Blodgett trough is a KSV NIMA Medium model (L: 364 × W: 75 × H: 4 mm^3^ purchased from Quantum Design SARL, Les Ulis, France) and is made of Teflon and equipped with two motorized Teflon barriers moving symmetrically at a constant speed. Surface pressure is measured using a calibrated Whilelmy blat connected to a force sensor with a resolution of 0.1 µN∙m^−1^. Volumes between 10 and 100 µL of solutions were spread at the air–water interface at a constant temperature of 20 °C using both a thermostated bath for the subphase and ambient air conditioning. Then, 10 min after solvent evaporation, the compression of the film was carried out at 6 mm∙min^−1^ and the surface pressure was plotted as a function of the mean area.

BAM images of all studied films were taken by the apparatus developed by Meunier et al. [25] and were recorded over the entire surface pressure range of the compression isotherm. When the incident beam was adjusted to the Brewster angle at around 53° and polarized parallel to the incidence plane, the reflected intensity was zero (beam is totally refracted) in the absence of a film over the water surface, whereas it was no longer zero in the presence of a film. This intensity was a function of the refractive index profile n(z) and layer thickness. The reflected beam was collected by an objective connected to a camera allowing the visualization of the surface in real time. The size of the BAM images was 600 × 600 µm with a lateral resolution of approximately 1 µm.

### 2.3. Synthesis

#### 2.3.1. Synthesis of TMe-βCD and TMe-γCD

The synthesis and characterizations of host TMe-βCD and TMe-γCD molecules were described in detail elsewhere [26,27].

#### 2.3.2. Synthesis of EDOT∙TMe-βCD and EDOT∙TMe-γCD

EDOT∙TMe-βCD and EDOT∙TMe-γCD starting monomers were prepared according to our recently reported procedure [28].

#### 2.3.3. The Synthesis of Py-EDOT-Py

EDOT (0.1 mL, 0.143 g, 1.0 mmoL) was dispersed in ultrapure water (5.0 mL) by sonication for 15 min followed by vigorous stirring at ambient temperature. Py (0.446 g, 2.2 mmoL) dissolved in acetone (15.0 mL) was added to the EDOT aqueous dispersion. FeCl_3_ (0.615 g, 3.8 mmoL) was added to the solution and then vigorously stirred at ambient temperature for 24 h in the dark under N_2_ protection. The resulting solid was filtered and washed with water and ethanol and dried in a vacuum oven at 90 °C for 24 h. After being dried, 0.118 g as a dark solid in a 20.4% was obtained. 

^1^H-NMR (400 MHz, CDCl_3_, δ, ppm): 8.54–8.09 ppm (H, Py), 4.68–4.08 ppm (H, CH_2_ from EDOT).

#### 2.3.4. The Synthesis of Py-EDOT∙TMe-βCD-Py

EDOT (0.1 mL, 0.143 g, 1.0 mmoL) was added to an aqueous solution of TMe-βCD (1.454 g, 1.02 mmoL) in ultrapure water (8 mL). The dispersion was sonicated for 15 min. After 10 min, the solution became opalescent and then a semisolid product was deposited in the flask. The obtained semisolid product was then vigorous stirred at ambient temperature for 24 h in the dark under N_2_ protection. Py (0.446 g, 2.2 mmoL) solubilized in 15 mL of acetone and in one portion FeCl_3_ (0.81 g, 5.0 mmoL) was added to the suspension. A stream of N_2_ was bubbled through the mixture to remove dissolved oxygen and the flask was capped and protected against light. The reaction mixture was stirred for 4 days, during which a green suspension was obtained. The solid was isolated by filtration, and finally thoroughly washed with water and ethanol. After drying in a vacuum oven at 90 °C for 24 h, 0.589 g as a dark-green solid in a 28.83% yield was obtained. The resulting compound was dissolved in CHCl_3_ and the soluble part was collected. The soluble fraction in CHCl_3_ was concentrated by vacuum evaporation and precipitated in diethyl ether. The precipitate was filtered and dried. After being dried, 0.108 g, 18.3% yield (calculated from the obtained 0.589 g product) as a dark-green solid was obtained.

^1^H-NMR (400 MHz, CDCl_3_, δ, ppm): 8.53–8.03 ppm (H, Py), 5.13 ppm (7H, H1, TMe-βCD), 4.69–4.39 (H, CH_2_ from: Py-EDOT, Py-EDOT-Py and Py-EDOT-EDOT), 3.79–3.20 (H2-6, -CH_3_ from 2, 3, and 6 positions of TMe-βCD).

#### 2.3.5. The Synthesis of PEDOT∙TMe-βCD and PEDOT∙TMe-γCD PPs

PEDOT∙TMe-βCD and PEDOT∙TMe-γCD PPs were prepared according to previously reported procedures [22].

^1^H-NMR (400 MHz, DMSO-d_6_, δ, ppm): 5.05 ppm (d, H1, TMe-βCD), 4.43 ppm (CH_2_ from PEDOT), 3.72–3.01 (H2-6, -CH_3_ from 2, 3, and 6 positions of TMe-βCD overlapped with water from DMSO-d_6_).

#### 2.3.6. The Synthesis of PEDOT∙TMe-βCD and PEDOT∙TMe-γCD PRs

PEDOT∙TMe-βCD and PEDOT∙TMe-γCD PRs were synthesized by similar experimental conditions as previously reported [22], except that 109 mg of Py as stopper instead of triphenylmethane dissolved in 10 mL of acetone and 0.107 g FeCl_3_ as fresh catalyst was added at the end of the polymerization processes and the reactions were continued for two days. The resulting solids were filtered and washed with water, methanol, and acetone in succession and dried. After drying, the resulting dark-green solids were dispersed in CHCl_3_, vortex stirred for 15 min, and their insoluble parts isolated by filtration. The soluble fractions in CHCl_3_ were transferred into a round bottom flask, concentrated by rotary evaporation and precipitated in heptane. The precipitates were collected, dried, and chemically characterized.

^1^H-NMR (400 MHz, DMSO-d_6_, δ, ppm): 8.33–8.30 ppm (H, Py), 5.05–5.04 ppm (H1, TMe-βCD), 4.25–4.17 (CH_2_, PEDOT), 3.73-3.06 (H2-6, -CH_3_ from 2, 3, and 6 positions of TMe-βCD overlapped with water from DMSO-d_6_).

## 3. Results and Discussion

### 3.1. Synthesis and Characterization 

The preparation of PEDOT∙TMe-βCD and PEDOT∙TMe-γCD involved as a first step the threading of the neutral EDOT guest into TMe-βCD or TMe-γCD cavities, thus leading to EDOT∙TMe-βCD and EDOT∙TMe-γCD encapsulated monomers. The synthesis of TMe-βCD and TMe-γCD macrocycles were performed according to previously reported procedure [26]. The ^1^H-NMR and ^13^C-NMR data confirmed the chemical structure of TMe-βCD and TMe-γCD host molecules [27]. Further, TMe-βCD and TMe-γCD macrocycles solubilized in a minimal amount of water were subjected to encapsulating the EDOT into their cavities based on intermolecular interactions, thus leading to EDOT∙TMe-βCD and EDOT∙TMe-γCD [28]. The ability of TMe-βCD or TMe-γCD host molecules to bind the neutral guest EDOT was demonstrated by UV–vis titrations in water. The constant stability (*K_s_*) values were found to be ~1.0 × 10^3^ M^−1^, thus confirming the ability of macrocyclic molecules to bind the EDOT monomer inside of their cavity. Furthermore, the binding ability of the EDOT starting monomer to the hosts TMe-βCD and TMe-γCD was verified by molecular docking simulation. The results of computational data reveal the occurrence of hydrophobic interactions, which contribute to the insertion of the EDOT guest inside the macrocyclic cavities and a better binding of the EDOT to TMe-βCD [28]. The ^1^H-NMR spectra of the EDOT∙TMe-βCD and EDOT∙TMe-γCD denoted chemical shift displacements for all the protons of EDOT and macrocyclic molecules. According to the ^1^H-NMR analysis, the EDOT was included in the central cavities of TMe-βCD and TMe-γCD owing to the largest chemical shift variations that were identified for the H-3 and H-5 protons located inside the macrocyclic cavity [28]. In order to achieve a more detailed structural assessment of the EDOT∙TMe-βCD and EDOT∙TMe-γCD compounds, two-dimensional H-H ROESY analyses were further performed. The H-H ROESY spectra showed correlation peaks between H-3 and H-5 of TMe-βCD or TMe-γCD and the aromatic protons of EDOT. In comparison, no correlation peaks between EDOT and protons from the outside cavity (H-1, H-2, H-4) of TMe-βCD or TMe-γCD were detected. The H-H ROESY suggested that the EDOT molecule was included with the thiophene ring deep inside the macrocyclic cavities. Several chemical shift displacements of the protons from the outside cavities of macrocycles, and the -OMe groups were ascribed to conformational changes of the glucopyranose units that occur upon EDOT complexation [29,30]. To further support the presence of TMe-βCD or TMe-γCD on the EDOT backbones, the MALDI TOF MS associated to laser-induced dissociation fragmentation (LID MS/MS) in a TOF/TOF MALDI MS setup, operated in LIFT mode and TGA analysis clearly provided evidence of EDOT encapsulation inside the macrocyclic cavities. After these wide characterizations, the resulting EDOT∙TMe-βCD and EDOT∙TMe-γCD were subjected to an oxidative coupling reaction in water using FeCl_3_ when PEDOT∙TMe-βCD and PEDOT∙TMe-γCD in PP architectures were obtained. Afterwards the attachment of bulky Py groups afforded the synthesis of PEDOT∙TMe-βCD and PEDOT∙TMe-γCD PRs. Finally, only their soluble fractions in CHCl_3_ were selected and precipitated in heptane. The use of TMe-βCD or TMe-γCD host molecules led to compounds soluble in DCM, THF, and DMSO, and can form stable colloidal dispersions in water. In order to evidence the presence of the Py group at both end-chains of PEDOT, a difunctionalized EDOT (Py-EDOT-Py) and its encapsulated form (Py-EDOT∙TMe-βCD-Py) were synthesized and chemically characterized (see above in Materials and Methods). Using ^1^H-NMR in CDCl_3_, the presence of both Py ends in the Py-EDOT-Py and Py-EDOT∙TMe-βCD-Py were confirmed (Appendix A). These conclusions are further supported by MALDI MS/MS where the appearance of a monoisotopic peak with m/z = 544.1 in the MS spectrum confirms the structural assignment of Py-EDOT-Py+ ion species (Appendix A). The chemical structures of the resulting PEDOT∙TMe-βCD and PEDOT∙TMe-γCD compounds were validated by using FT-IR (Appendix A) and NMR spectroscopy. The FT-IR spectra show all of the characteristic bands of PEDOT at 1323, 1157, 980, 921, 817, and 673 cm^−1^ [31], and additional bands located at 1073, 1060, 571, 519, and 434 cm^−1^ that evidence the presence of the TMe-βCD or TMe-γCD [28,32]. The presence of characteristic vibration at 3043 cm^−1^ (C-H stretching mode), 1589 cm^−1^ (C=C stretching), 1157 cm^−1^ (C=C bending), 832 and 702 cm^−1^ (C-H bending) also evidenced that the Py stopper was attached to the PEDOT ends [33]. The ^1^H-NMR spectra in DMSO-d_6_ of PEDOT∙TMe-βCD and PEDOT∙TMe-γCD compounds exhibited all the characteristic protons. The ^1^H-NMR spectrum of PEDOT∙TMe-βCD shows a peak at 5.05 ppm corresponding to the anomeric proton of TMe-βCD and a broad peak at 4.51 ppm corresponding to methylene protons from the PEDOT. The rest of the peaks from TMe-βCD appearing at 3.73, 3.49, 3.24, and 3.06 ppm are partially overlapped with the water from solvent (Appendix A). The peaks at 8.33, 8.31, and 8.21 ppm in NMR spectrum of PEDOT∙TMe-βCD are also indicative of the presence of the bulky Py units. Nevertheless, the ^1^H-NMR spectra of both compounds show broad peaks associated to π-π interactions in DMSO-d_6_ solutions [34,35]. By using the ratio of the integrated area of the H-1 from TMe-βCD (5.05 ppm, I_H-1_) and the methylene proton peaks of the PEDOT (4.25 ppm, I_PEDOT_) (_IH-1_/7)/(I_PEDOT_/4) the molar ratio in PEDOT∙TMe-βCD was found to be ~ 40.0%. In comparison, the ^1^H-NMR spectrum of PEDOT∙TMe-γCD (Appendix A) indicated a lower molar ratio ~30.0 %. As a result of the high molar ratio of TMe-βCD, the PEDOT∙Tme-βCD exhibited better film-forming ability on different substrates. 

The molecular weight distribution (*M_n_*) and polydispersity index (*M_w_*/*M_n_*) of PEDOT PEDOT∙Tme-βCD and PEDOT∙Tme-γCD estimated by GPC are summarized in Table 1. The *M_n_* values of PEDOT∙Tme-βCD and PEDOT∙Tme-γCD are higher than those of the PEDOT. As expected, a unimodal distribution of the investigated compounds denotes no dethreading of Tme-βCD or Tme-γCD from the PEDOT backbones (Appendix A). The higher *M_w_*/*M_n_* value of PEDOT∙Tme-βCD and PEDOT∙Tme-γCD is presumably due to a compositional polydispersity of the PEDOT chains. 

It should be pointed out that the *M_n_* values estimated by GPC analysis have to be taken as indicative only. The difference between the rigid rod-like structures of the investigated compounds cannot be well correlated with the coil-like structure of Pst standards.

### 3.2. Thermal Analysis

Appendix A depicted the TGA analyses of PEDOT∙TMe-βCD and PEDOT∙TMe-γCD and those of the reference PEDOT and the data are summarized in Table 2. The TGA results indicated that the PEDOT polymer undergoes three thermal degradation steps starting from 187 °C, like previously results [36]. Upon encapsulation of the PEDOT into the TMe-βCD cavity, the TGA curve exhibited only one thermal decomposition step with an increased T_onset_ to about 207 °C. It should be noted that the TMe-βCD macrocycle is a stable compound with T_onset_ = 318 °C [28], which is higher than the first degradation step of the PEDOT∙TMe-βCD. This observation indicates that the formation of PEDOT∙TMe-βCD decreases the macrocycle backbone stability.

Further insight is provided by DSC analysis (Appendix A). The DSC curves of the investigated compounds indicated no endothermic peaks corresponding to glass transition or melting temperatures in the range 0–200 °C, which was also found for the PEDOT encapsulated by cucurbit [7]uril [36].

### 3.3. Optical Properties

The optical properties of PEDOT∙TMe-βCD and PEDOT∙TMe-γCD were investigated in H_2_O and ACN solutions and the results are summarized in Table 3. In H_2_O, the absorption spectra of PEDOT∙TMe-βCD and PEDOT∙TMe-γCD are broad and featureless (Appendix A). Appendix A illustrates the absorption in ACN, where several bands appeared in the 300–400 nm interval which were assigned to the π-π* and n-π* transitions [37]. F_L_ spectra of PEDOT∙TMe-βCD and PEDOT∙TMe-γCD (λ_ex_ = 375 nm) exhibited different bands and shoulders in H_2_O (Appendix A). F_L_ spectra at different excitation wavelengths in ACN (Appendix A) are dependent on the excitation wavelength denoting the existence of a distribution of energetically different molecules in the ground state coupled with a low rate of the excited state relaxation processes [38]. The fluorescence lifetime (τ) of PEDOT∙TMe-βCD and PEDOT∙TMe-γCD (Table 3) in both H_2_O and ACN exhibited bi-exponential decay traces, which were assigned to the contribution of intrachain emission and excitonic contribution [38]. We also note that the PEDOT∙TMe-βCD and PEDOT∙TMe-γCD exhibited both Φ_PH_ and Φ_FL_ in ACN, whereas in H_2_O these efficiencies are too low, evidencing mainly nonradiative pathways [16,39].

Furthermore, the nanosecond transient absorptions (nsTA) in H_2_O and ACN were also performed (Figure 2). The nsTA map of PEDOT∙TMe-βCD in H_2_O reveals at 290 nm a ground state bleaching band (GSB) (Figure 2a). At shorter wavelengths (210, 230, and 250 nm) appear some excited state absorption bands (ESA) and more than one excited state (S_n_ > 1). The negative bands assigned to stimulated emissions (SE) appeared at 395 and 403 nm longer wavelengths. In ACN (Figure 2b), the GSB band appeared at 306 nm, the ESA at 225 and 250 nm, and more than one excited state (S_n_ > 1). At 391 and 440 nm wavelength, two negative bands appeared assigned to the SE that can be a result of the triplet manifold, confirming the P_H_ properties of the PEDOT∙TMe-βCD compound.

Figure 3a shows the nsTA map of PEDOT∙TMe-γCD in H_2_O where the GBS band appeared at 295 nm, the ESA band at 215 and 240 nm, and negative bands that are assigned to the SE and more than one excited state (S_n_ > 1) at 404 nm. Figure 3b displays the nsTA map of PEDOT∙TMe-γCD in ACN, when the GBS bands appeared at 270 and 315 nm, the ESA bands at 215 and 240 nm, and more than one excited state (S_n_ > 1). At longer wavelengths 390, 440, and 455 nm, the presence of SE negative bands ascribed to the triplet manifold also confirms its P_H_ properties.

The optical results indicated that PEDOT∙TMe-βCD and PEDOT∙TMe-γCD are sensitive to the polarity changes of the microenvironment, in agreement with previous reported results [39]. Based on optical investigations, it is perceivable that these encapsulated PEDOT compounds exhibited better P_H_ and increased Φ_FL_ and Φ_PH_ efficiencies in ACN than in H_2_O.

### 3.4. Electrochemical Properties

The redox properties were also investigated by CV (Appendix A). The CV curves of PEDOT∙TMe-βCD and PEDOT∙TMe-γCD films display an electroactive response, which corresponds to the positive doping of PEDOT. It should be pointed out that the redox behaviors of the investigated compounds have a similar origin with those previously reported [22,39]. According to the electrochemical results, it can be concluded that the investigated PEDOT∙TMe-βCD and PEDOT∙TMe-γCD exhibit typical insulating behavior in a wide range of potential between n- and p-doping processes, denoting their semi-conducting properties.

### 3.5. Surface Morphology

Figure 4 depicts the surface topography of PEDOT∙TMe-βCD and PEDOT∙TMe-γCD films prepared by drop casting and dried at ambient temperature from H_2_O, ACN, and THF solutions. The features of AFM image are strongly influenced by the solvent polarity. For both samples, the H_2_O induces agglomeration and the shape of grains became irregular [36,39]. The AFM image of the PEDOT∙TMe-βCD film obtained from a dilute ACN solution appeared as isolated grains with mean diameters of 160 ± 31 nm, whereas the PEDOT∙TMe-γCD presented individual entities, spherically shaped grains with two different diameters 217  ±  26 and 84  ±  12 nm. The AFM analysis showed obvious differences between the surface morphologies of the thin films obtained from a dilute THF solution. The AFM image of both PEDOT∙TMe-βCD and PEDOT∙TMe-γCD films obtained from THF appeared as individual, small grains with an average particle size of 67 ± 11 nm and 79 ± 21 nm, respectively. The AFM results suggest that the packing geometries of PEDOT∙TMe-βCD and PEDOT∙TMe-γCD are not different but are clearly induced by the nature of solvent, in good agreement with the optical data.

### 3.6. Dynamic Light Scattering (DLS)

The dynamics at molecular level of the water stable dispersion of PEDOT∙TMe-βCD and PEDOT∙TMe-γCD were investigated by DLS technique. The PEDOT∙TMe-βCD exhibited a sharp monodisperse peak with DH and ZP-ζ of 55 ± 15 nm and −31.23 mV, respectively, (Figure 5). The ZP-ζ value of −31.23 mV denotes that the PEDOT∙TMe-βCD colloidal system became extremely stable [11,40]. Another important aspect that should be noted is that in terms of DLS results, the PEDOT∙TMe-βCD revealed a better distribution as was found for the PEDOT with anionic poly(styrenesulfonate) [40]. At the same time, the water dispersion of PEDOT∙TMe-γCD reveals a high DH value of (122 ± 32 nm) and smaller ZP-ζ value (−20.38 mV). This behavior may be associated with an increased tendency of its colloidal particles to aggregate. Additionally, the values of ZP-ζ denote that a negatively charged layer covers on the surfaces of the PEDOT∙TMe-βCD and PEDOT∙TMe-γCD colloidal particles. The conclusion drawn from the DLS approach is that the PEDOT∙TMe-βCD is a more stable colloidal dispersion in water than those of PEDOT∙TMe-γCD due to higher electrostatic repulsion, which prevents particle agglomeration. It should be note that the lower molar ratio of the PEDOT∙TMe-γCD compound provides evidence of the changes in its water dispersion stability.

### 3.7. Electrical Properties

With a view to perceiving the effect of TMe-βCD and TMe-γCD encapsulation on the electrical conductivity of PEDOT∙TMe-βCD and PEDOT∙TMe-γCD, the broadband dielectric spectroscopy (BDS) was carried out. Figure 6 shows the evolution of measured conductivity (σ) with frequency (ƒ) and the phase angle (*θ*) at different temperatures of PEDOT∙TMe-βCD and PEDOT∙TMe-γCD. The measured *σ* of the materials was calculated according to Equation (1):(1)$$\sigma =2\pi f\varepsilon \prime \prime \;\varepsilon_{0}$$
where ƒ is the electrical field frequency, *ε*″ represents the dielectric loss, and *ε*_0_ is the vacuum permittivity.

The *θ* = tan^−1^(Z_im_/Z_re_), where Z_im_ and Z_re_ are imaginary and real components of the impedance.

It is important to note that taking the ƒ window between 1 and 100 Hz, the values of *θ* are close to zero and the measured σ is independent of ƒ. This behavior is generally attributed to the direct current conductivity (σ_DC_) and may be associated with the movement of free charge carriers through the PEDOT polymer backbone [41]. The measured σ increases continuously when the ƒ increases whereas the *θ* values are deviated from 0 °C (100 Hz) up to −70 °C (1 MHz) (Figure 6). The variation of σ and *θ* with ƒ is characteristic of alternating current conductivity (σ_AC_) and associated with the dipolar relaxation phenomenon, which occurs in the PEDOT polymer backbone [41]. It is noteworthy that the flat plateau region of σ_DC_ is detected even at low temperatures (−50 °C) and gradually extended to higher ƒ towards increasing temperature to the superior limit of 200 °C. This would suggest that the σ_DC_ of free charge carriers is a thermally activated process and may be treated as a relaxation-type phenomenon [42]. Furthermore, to have clear information on the σ_DC_ of PEDOT∙TMe-βCD and PEDOT∙TMe-γCD, their values were estimated from the plateau of the σ and *θ* spectral regions [42,43] (Figure 7).

It is worth noting that the numerical values of σ_DC_ increasing from ~10^−7^ S∙cm^−1^ at −50 °C (superior limit of insulators) to ~10^−5^ S∙cm^−1^ (inferior limit of semiconducting materials) at 200 °C (Table 4). The effect of thermal activation indicates superior values of σ_DC_ at 200 °C in PEDOT∙TMe-γCD than those of the PEDOT∙TMe-βCD sample. Such results are attributed to the thermally activated charge carrier’s conductivity. The values of activation energy (*E_a_*) of charge carriers in PEDOT∙TMe-βCD and PEDOT∙TMe-γCD were also evaluated by Arrhenius equation and the results are summarized in Table 4. The estimated *E_a_* values considered as the minimum amount of energy required for charge carrier movement through the material are in the range of meV, following similar trends to other CPs [44]. The lowest *E_a_* value of PEDOT∙TMe-βCD reveals its better transport of electrons between active sites than those of PEDOT∙TMe-γCD.

Furthermore, the σ_DC_ value against the inverse of temperature was also investigated (Appendix A). In the temperature range between 0 and 70 °C, the σ_DC_ is quite linear and decreases with the reciprocal of absolute temperature. Generally, a typical evolution of σ_DC_ towards the inverse of temperature of CPs follows a linear dependency as an Arrhenius-type behavior [45].

### 3.8. 2D Supramolecular Organizations at the Air–Water Interface

To gain further insight into the effect of macrocyclic encapsulations, the supramolecular arrangements of thin films produced at air–water interfaces were further investigated. Figure 8A shows compression isotherms measured for the starting EDOT monomer depending on whether TMe-βCD macrocycle and Py end-groups are present. These isotherms are presented as a function of the mean area per EDOT to better highlight the contribution of the TMe-βCD macrocycle and Py blocking groups. Figure 8A (dashed curve) shows that Py-EDOT-Py does not form any Langmuir monolayer since the surface pressure does not rise due to probable dissolution in the water subphase.

In the EDOT∙TMe-βCD inclusion complex, the behavior strongly depends on whether or not Py blocking groups are present. Figure 8C (blue frame) of BAM analysis for EDOT∙TMe-βCD shows that at areas per EDOT below 70 Å², the surface pressure increases continuously until 8 mN∙m^−1^ and the monolayer appears in a very homogeneous condensed state. As can be seen from the BAM image, the formation of small bright aggregates reflecting the collapse of the monolayer is also present. Figure 8A (black curve) indicates a different behavior for Py-EDOT∙TMe-βCD-Py, where the surface pressure rises to significantly higher values (~60 mN∙m^−1^) indicating a strong increase of the film stability. The surface pressure slowly increases before 8 mN∙m^−1^ and then sharply. As can be seen from the BAM analysis, in Figure 8C (black frame), the film appears homogeneous at non zero surface pressure until a curve inflection is observed at around 60 mN∙m^−1^ where 3D aggregates are identified. More than that, the film becomes rigid beyond 8 mN∙m^−1^. In comparison with the EDOT∙TMe-βCD isotherm, the slope break at ~8 mN∙m^−1^ may correspond to a transition towards vertically aligned macrocycles with respect to the air–water interface. It should be pointed out that below 8 mN∙m^−1^, the monolayers for both compounds are highly compressible. These results can be attributed to the macrocycles progressively straight up upon compression. By extrapolation of the slope of the isotherm in the condensed phase to zero surface pressure, the mean area per EDOT is found to be 15 Å² compared with 50 Å² for EDOT∙TMeβCD. This difference could be related to a better order of the macrocycles within the Py-EDOT∙TMe-βCD-Py even in the expanded state, as a consequence of the Py group anchorage at the air–water interface. With a view to understanding the factors that control these isotherms, the plotted isotherms of EDOT∙TMe-βCD and Py-EDOT∙TMe-βCD-Py as a function of the mean area per TMe-βCD macrocycle were compared to those of the TMe-βCD isotherm (Figure 8B). As expected, the shape of the isotherms and morphology identified by BAM analysis were similar for EDOT∙TMe-βCD and TMe-βCD. Taking into consideration that Py-EDOT-Py does not form any Langmuir monolayer, the identified shift towards higher molecular area for the EDOT∙TMe-βCD inclusion complex confirms the presence of the EDOT monomer into TMe-βCD cavities in the 2D supramolecular arrangement. However, the presence of the Py blocking groups in the Py-EDOT∙TMe-βCD-Py leads to a significant shift of the area per macrocycle towards lower areas. Indeed, by extrapolating the slope of the isotherms in the condensed state to zero surface pressure, limiting mean areas of 415 Å², 486 Å² and 140 Å² are obtained for TMe-βCD, EDOT∙TMe-βCD, and Py-EDOT∙TMe-βCD-Py, respectively (Figure 8B). Considering the internal and external diameters of TME-βCD, the macrocycle lying flat on the water surface would occupy at least a surface of 184 Å² [46]. Therefore, the TME-βCD and EDOT∙TMe-βCD form disordered monolayers, whereas the Py-EDOT∙TMe-βCD-Py leads to a more condensed phase with the macrocyclic molecule standing more vertically which can be attributed to the presence of Py end groups with beneficial effect on the 2D organization.

We found that the presence of the blocking Py groups improves both the organization of the ultrathin film but also the monolayer stability, since the collapse pressure which marks the loss of 2D arrangement is significantly increased. Figure 9A shows the compression isotherm of PEDOT∙TMe-βCD as function of the mean area per EDOT monomer and compared with those of Py-EDOT∙TMe-βCD-Py. We identified the similarity of both shape isotherms following a gradual increase in the surface pressure without any phase transition until at very high surface pressure (55–60 mN∙m^−1^) when the collapse appeared. It should be noted that the shift toward lower mean area per EDOT appeared only in the case of the PEDOT∙TMe-βCD monolayers. The mean area per EDOT in the condensed phase of Py-EDOT∙TMe-βCD-Py and EDOT∙TMe-βCD were found to be 15.0 Å² and 4.2 Å², respectively. Figure 9B and C show the effect of TMe-βCD or TMe-γCD macrocyclic encapsulation on the behavior of PEDOT∙TMe-βCD and PEDOT∙TMe-γCD. The shift toward lower mean area per EDOT can be associated with low molar ratio of TMe-γCD on the PEDOT backbones and the increased contribution of Py end-groups. As shown in Figure 9B (blue curve) of PEDOT∙TMe-βCD and of PEDOT∙TMe-γCD, Figure 9B (red curve), both isotherms exhibited very similar shapes. The BAM image of PEDOT∙TMe-βCD indicates the formation of homogeneous monolayers in the condensed phase, Figure 9C (blue frame), whereas in the PEDOT∙TMe-γCD monolayer, Figure 9C (red frame), some micrometer 3D aggregates are visible in the condensed state. The obtained BAM results prove the existence of collapses in both PEDOT∙TMe-βCD and PEDOT∙TMe-γCD monolayers ~55 mN∙m^−1^, which correspond to the formation of 3D aggregates. The mean area per EDOT measured in the condensed phase is 4.2 Å² and 4.4 Å² for PEDOT∙TMe-βCD and PEDOT∙TMe-γCD, respectively. These close values can be correlated with the larger size of the TMe-γCD macrocycle and lower molar ratio (30% instead of 40%) [28,46]. The obtained results strongly suggest that the presence of TMe-CDs on the PEDOT backbones and the presence of Py ends play an important role in the supramolecular arrangements of PEDOT∙TMe-βCD and PEDOT∙TMe-γCD layers. According to high surface pressure values reached during the film compression, we can conclude that PEDOT∙TMe-βCD and PEDOT∙TMe-γCD monolayers have improved stability at the air–water interface. These studies conclude that the monolayer properties, such as compressibility, mesoscopic-scale organization, and 2D stability can be controlled during the synthesis through the size of the macrocycle, the molar ratio, and the presence of blocking groups.

## 4. Conclusions

PEDOT∙TMe-βCD and PEDOT∙TMe-γCD were synthesized and photophysical characterized. The synthesis of such supramolecular compounds leads to distinct improvements with regard to the solubility in common organic solvents, the film forming ability, and the glass-transition temperatures, and it also plays an important role in the supramolecular organizations of PEDOT backbones. The threading of TMe-βCD or TMe-γCD into the PEDOT backbones provides an efficient strategy to enhance the photophysical properties of PEDOT encapsulated compounds. These compounds exhibited better photophysical characteristics, which were consistent with the electrochemical and surface morphological data. According to the AFM results, the packing geometries of PEDOT∙TMe-βCD and PEDOT∙TMe-γCD are not different but are induced by the nature of solvent. The DLS results evidenced that the water-soluble part of PEDOT∙TMe-βCD is more stable colloidal dispersion. It was found that both PEDOT∙TMe-βCD and PEDOT∙TMe-γCD could form 2D stable monolayers at the air–water interface. The present study is informative for estimating the impact of TMe-CDs encapsulation on the generation of PEDOT Langmuir monolayers. Based on these investigations, it is perceivable that the most important characteristics of PEDOT∙TMe-βCD and PEDOT∙TMe-γCD with respect to the non-rotaxane PEDOT are their enhanced solubility in organic solvents and improved photophysical properties, which definitely deserve attention as active layers in organic electronic devices. We are currently investigating this promising avenue.

## Figures and Tables

**Figure 1 materials-16-04757-f001:**
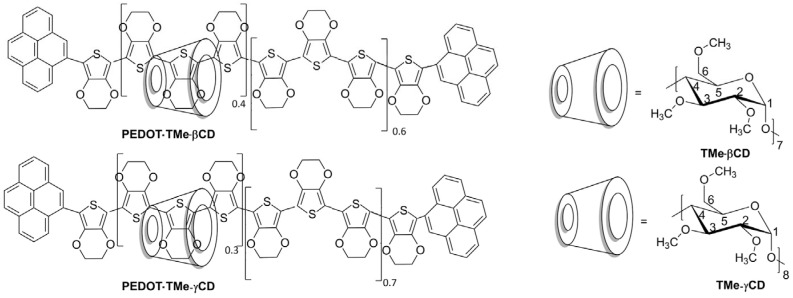
Chemical structures of PEDOT∙TMe-βCD and PEDOT∙Tme-γCD compounds.

**Figure 2 materials-16-04757-f002:**
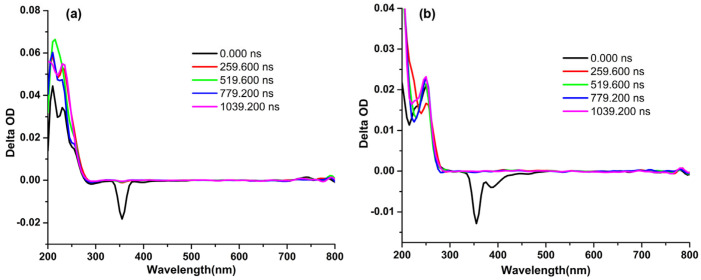
Nanosecond transient absorption of PEDOT∙TMe-βCD in H_2_O (λ_ex_ = 375 nm) (**a**) and ACN (λ_ex_ = 355 nm) (**b**).

**Figure 3 materials-16-04757-f003:**
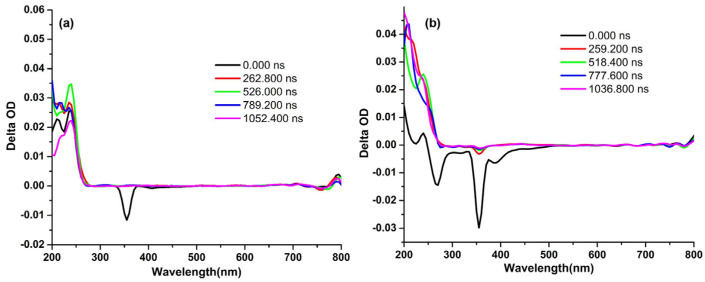
Nanosecond transient absorption of PEDOT∙TMe-γCD in H_2_O (λ_ex_ = 375 nm) (**a**) and ACN (λ_ex_ = 355 nm) (**b**).

**Figure 4 materials-16-04757-f004:**
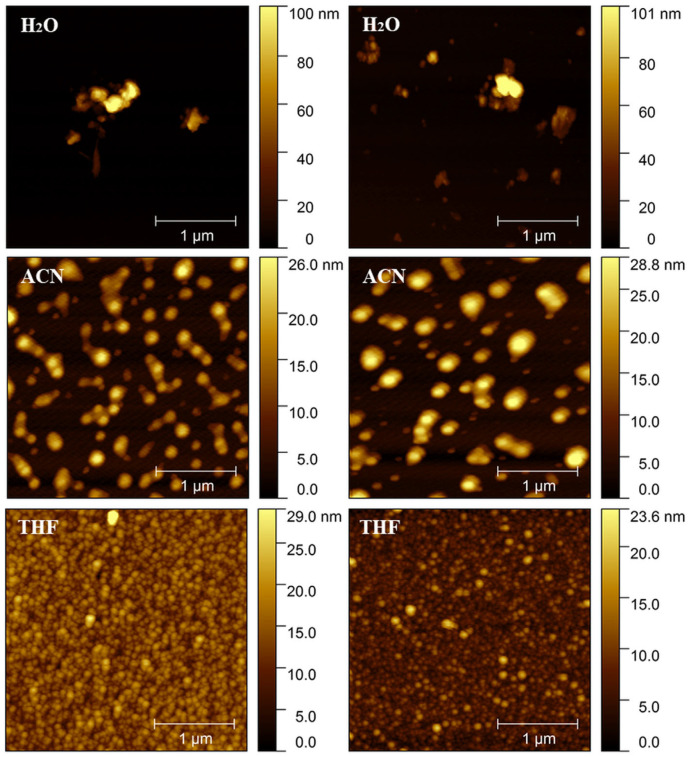
The AFM surface morphology over area of 3 × 3 µm^2^ of PEDOT∙TMe-βCD (**left**) and PEDOT∙TMe-γCD (**right**) films obtained by drop casting from H_2_O, ACN, and THF.

**Figure 5 materials-16-04757-f005:**
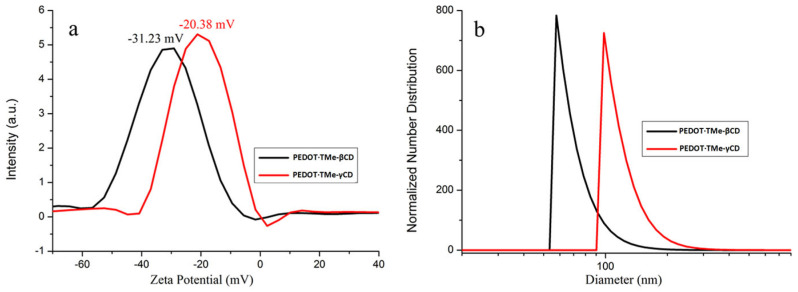
Zeta potential (**a**) and hydrodynamic diameter of PEDOT∙TMe-βCD and PEDOT∙TMe-γCD (**b**).

**Figure 6 materials-16-04757-f006:**
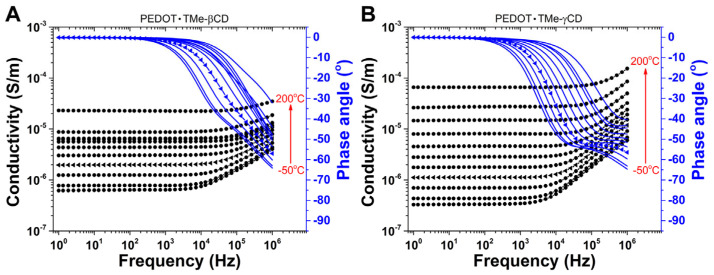
The variation of σ and *θ* with *f* at different temperatures for PEDOT∙TMe-βCD (**A**) and PEDOT∙TMe-γCD (**B**). Particularly for the temperature of 25 °C, the dielectric spectra are represented with solid triangle-type symbols.

**Figure 7 materials-16-04757-f007:**
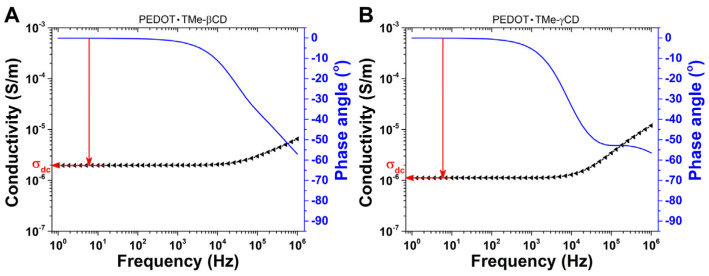
The σ_DC_ evaluation from the spectrum of PEDOT∙TMe-βCD (**A**) and PEDOT∙TMe-γCD (**B**) at 25 °C. The horizontal arrow illustrates the plateau region of the measured σ, while the vertical arrow shows the maximum value of the *θ*.

**Figure 8 materials-16-04757-f008:**
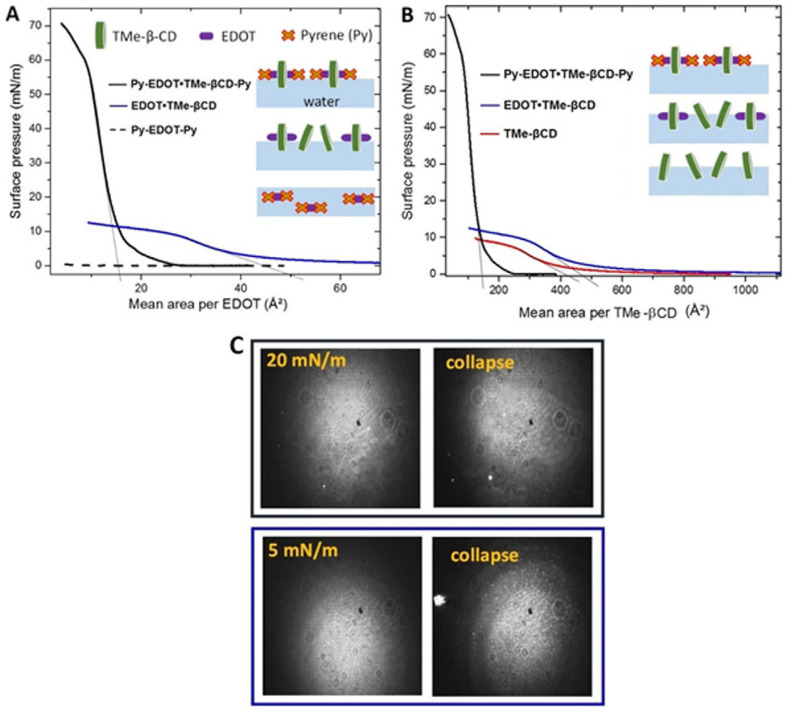
(**A**) Surface pressure–area isotherms for Langmuir films of Py-EDOT∙TMe-βCD-Py (black curve), EDOT∙TMe-βCD (blue curve) and Py-EDOT-Py (dashed curve) presented as a function of the mean area per EDOT monomer; (**B**) Py-TMe-βCD-Py (black curve), EDOT∙TMe-βCD (blue curve), and TMe-βCD (red curve) presented as a function of the mean area per TMe-βCD macrocycle. (**C**) BAM images (600 µm × 600 µm) of Py-EDOT∙TMe-βCD-Py (black frame) and EDOT∙TMe-βCD (blue frame) Langmuir films in the condensed and collapse phases.

**Figure 9 materials-16-04757-f009:**
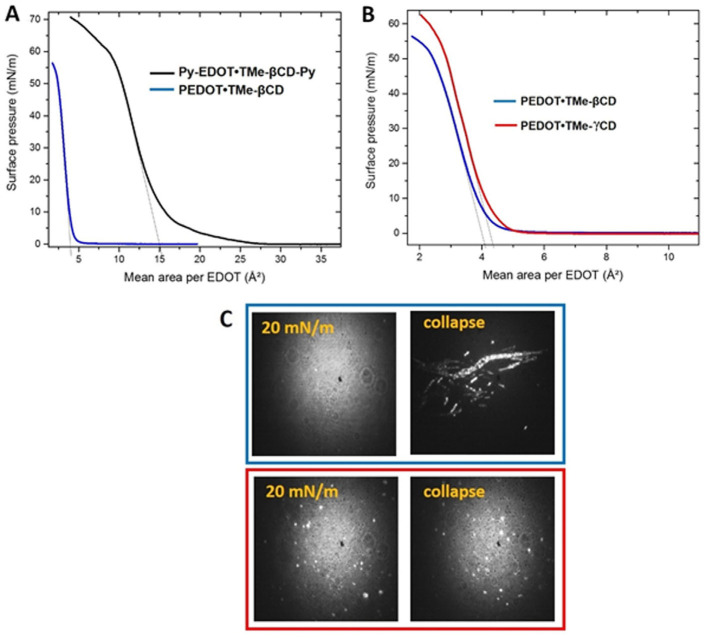
Surface pressure–area isotherms presented as a function of the mean area per EDOT monomer for: (**A**) Py-EDOT∙TMe-βCD-Py (black curve) and PEDOT∙TMe-βCD (blue curve); (**B**) PEDOT∙TMe-βCD and PEDOT∙TMe-γCD (red curve) and (**C**) BAM images (600 µm x 600 µm) of PEDOT∙TMe-βCD (blue frame) and PEDOT∙TMeγ-CD (red frame) Langmuir films in the condensed phase and at the collapse.

**Table 1 materials-16-04757-t001:** Molecular weight (*M_n_*), polydispersity index (*M_w_*/*M_n_*) of PEDOT, PEDOT∙Tme-βCD, and PEDOT∙TMe-γCD compounds.

Sample	*M_n_* (g∙moL^−1^)	*M_w_*/*M_n_*
PEDOT	700	1.10
PEDOT∙TMe-βCD	15,000	1.37
PEDOT∙TMe-γCD	13,000	1.49

**Table 2 materials-16-04757-t002:** Thermogravimetric data of the compounds.

Sample	Step	T_onset_ ^(a)^ (°C)	T_peak_ ^(b)^ (°C)	T_endset_ ^(c)^ (°C)	W ^(d)^ (%)	Residue ^(e)^ (%)
PEDOT	I	187	216	265	4.34	
	II	308	367	399	40.41	37.13
	III	670	698	745	18.12	
PEDOT∙TMe-βCD	I	207	374	418	55.00	45.00

^(a)^ The start temperature of the degradation process. ^(b)^ The maximum degradation temperature. ^(c)^ The temperature of complete degradation process. ^(d)^ The mass percentage loss recorded at each stage. ^(e)^ The residue at the end of degradation process.

**Table 3 materials-16-04757-t003:** Emission lifetimes (τ), quantum yields for fluorescence (Φ_FL_), and phosphorescence (Φ_PH_) determined by nanosecond transient absorption of PEDOT∙TMe-βCD and PEDOT∙Tme-γCD.

Sample	Solvent	λ_ex_ (nm)	λ_em_ (nm)	τ_1_ (ns)	τ_2_ (ns)	Φ_FL_ (%)	Φ_PH_ (%)
PEDOT∙Tme-βCD	H_2_O	375	403	5.498 (66.79%)	28.947(33.21%)	2.05	0.05
PEDOT∙Tme-γCD	“	“	404	1.766 (29.31%)	9.706 (70.69%)	2.19	0.10
PEDOT∙Tme-βCD	can	355	391	1134 (69.80%)	8937 (30.20%)	4.89	76.17
PEDOt∙TMe-γCD	“	“	390	1.377 (21.88%)	10.492 (78.12%)	2.87	47.22

**Table 4 materials-16-04757-t004:** Numerical values of σ_DC_ at different temperature range and the *E_a_* of σ-relaxation process for PEDOT∙TMe-βCD and PEDOT∙TMe-γCD.

Sample	−50 °C	σ_DC_ (S∙cm^−1^) 25 °C	200 °C	*E_a_* (meV)
PEDOT∙TMe-βCD	6.2 × 10^−7^	2.0 × 10^−6^	2.3 × 10^−5^	138
PEDOT∙TMe-γCD	3.3 × 10^−7^	1.2 × 10^−6^	6.6 × 10^−5^	148

## Data Availability

The authors confirm that the data supporting the findings of this study are available within the article.

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
