# Peer review of "Novel Insight into the Photophysical Properties and 2D Supramolecular Organization of Poly(3,4-ethylenedioxythiophene)/Permodified Cyclodextrins Polyrotaxanes at the Air–Water Interface"

_materials, 2023, doi:10.3390/ma16134757_

Round 1
Reviewer 1 Report
Organic semiconductor, a kind of soft functional material exhibiting great potential for optoelectronics application, has been obtaining more attention recently. For selecting such organic semiconductor, the poly (3,4-ethylenedioxythiophene) polyrotaxanes end-capped by pyrene were prepared by oxidative polymerization of EDOT in this study. Two encapsulated PEDOT supramolecular materials (PEDOT∙TMe-βCD and PEDOT∙TMe-γCD) have been synthesized, with improved solubility and processability and thus enhanced thermal, optical, electronic properties. Detailed analysis results show that the encapsulated PEDOT compounds would form stable monolayers at the air-water interface after synthesis and imply possible application for optoelectronics. This submission is considerably prepared and might be helpful to the development of high performance organic semiconductor materials for a wide range of optoelectronics applications, based on the conjugated polymers.
For matching the publication requirements of MDPI journals, further modification of the manuscript is encouraged, and the followed suggestions are presented as comments. (1). Is it possible to modify the structures or surface morphology of the PEDOT∙TMe-βCD and PEDOT∙TMe-γCD after synthesis processing? (2) The relationship of enhanced photophysical properties of the encapsulated PEDOT compounds with the 2D supramolecular organizations at the air-water interface should explicate more clear in the manuscript. (3) For this study, how to control the 2D monolayer size and uniformity during synthesis of the PEDOT∙TMe-βCD and PEDOT∙TMe-γCD?
Author Response
We would like to thank the reviewer #1 for the constructive and valuable comments. The changes according to the reviewer' requirements have been included in the Main-Manuscript and they are highlighted in red.
Q1: Is it possible to modify the structures or surface morphology of the PEDOT∙TMe-βCD and PEDOT∙TMe-γCD after synthesis processing?
Answer Q1: We appreciate the validity of this question and we agree with the reviewer’s remark. Unfortunately, by using chemically permodified CDs derivatives as host molecules, the structures or surface morphology of conjugated polyrotaxanes cannot be further modify. All hydrogen atoms from hydroxyl groups of CDs that are susceptible for further modifications are substituted with methyl groups. We used these permodified derivatives for increasing the binding ability of the hydrophobic EDOT guest molecule and for improving the solubility in common organic solvents and photophysical properties of PEDOT polymer, as we report in this Ms. There is the possibility to modify the structures and surface morphology only by using native cyclodextrins or partially randomly modified derivatives. (Please, see Novel supramolecular networks based on PEG and PEDOT cross-linked polyrotaxanes as electrical conductive materials, DOI10.1016/j.eurpolymj.2019.02.015).
The related explanation has not been included in the revised manuscript.
Q2. The relationship of enhanced photophysical properties of the encapsulated PEDOT compounds with the 2D supramolecular organizations at the air-water interface should explicate more clearly in the manuscript.
Answer Q2: We agree with the reviewer’s remark and apologize for the lack of clarity. The question of the referee #1 comes from the sentence in the conclusion line 622. On this viewpoint, we have to mention that all photophysical characteristics were performed before the Langmuir monolayer formation. As a result, we cannot make correlations between these aspects. This point has been clarified in the new version of the main manuscript where we included some improvements in the 3.8 part and highlighted in red. The related explanation has not been included in the revised manuscript.
Q3. For this study, how to control the 2D monolayer size and uniformity during synthesis of the PEDOT∙TMe-βCD and PEDOT∙TMe-γCD?
Answer Q3: We agree with the reviewer’s remark and we give the answer to this comment at the end of 3.8 part accordingly:
These studies conclude that the monolayer properties, such as compressibility, mesoscopic-scale organization and 2D stability can be controlled during the synthesis through the size of the macrocycle, the coverage ratio and the presence of blocking groups.

Reviewer 2 Report
This paper describes the synthesis and properties of polyrotaxanes (PRs) comprising poly(3,4-ethylenedioxythiophene) (PEDOT) as an axle and permethylated b- or g-cyclodextrins (TMe-bCD or TMe-gCD) as a wheel. The synthesis of PRs was performed by the oxidative polymerization of inclusion complex consisting of EDOT and CD. The structures of PRs were analyzed by 1H NMR, IR, and GPC measurements. The properties of PRs such as thermal, optical, electrochemical, and electrical properties were evaluated and are discussed in the manuscript. Self-assembly behaviors and 2D supramolecular organizations at the air-water interface are also included in the manuscript.
The main problem of this manuscript is that the authors fail to provide the information enough for the validation of discussion. While the paper describes the detailed spectral assignments, the ROESY spectra, FT-IR spectra, 1H NMR spectrum of PEDOT-TMe-gCD in DMSO-d6 are lacking in the manuscript or the supporting information. I don’t agree with the calculation method for the coverage ratio of PEDOT-TMe-bCD. How many repeating units of PEDOT is sterically covered by a CD? The information should be absolutely included in the manuscript. Figure S3 indicates that the integral ratio between the anomeric protons and the methylene protons of PEDOT is 7 : 11.05. With the integral ratio, they concluded the coverage ratio to be 40%. If the numerical value of coverage ratio is correct, the results mean that one CD must sterically cover approximately 7 repeating unit of PEDOT. For example, it is well-known that an alpha-CD sterically covers two repeating units of PEG in the structure of PR comprising alpha-CD and PEG. Based on the ratio, rotaxane chemists calculate the coverage ratio of PRs as the ratio of sterically covered repeating units to the total repeating units of axle.
Considering the vague coverage ratio, I don’t agree with the sentence at line 315: Nevertheless, the 1H NMR spectra of both compounds show broad peaks associated to pai-pai interactions in DMSO-d6 solutions. Because the integral ratio in the 1H NMR spectrum seems to indicate the structure with high coverage ratio, the polymer would have no chance for intermolecular interactions.
The fluorescent spectra and the CV curves are also lacking, although the results are discussed in the manuscript. The authors should provide the data, otherwise omit the corresponding sentences from the manuscript.
At all, the manuscript is not acceptable at the present form, calling for the major revision.
Minor point:
The caption of Figure S7 repeatedly includes the word “absorption spectra”.
Author Response
We would like to thank the reviewer #2 for the constructive comments and suggestions. The changes according to the reviewers' requirements have been included in the Main-Manuscript or in the Supplementary Material, and they are highlighted in red. We believe the overall readability has been improved in this revised form of the manuscript, as recommended.
Q1: The main problem of this manuscript is that the authors fail to provide the information enough for the validation of discussion. While the paper describes the detailed spectral assignments, the ROESY spectra, FT-IR spectra, 1H NMR spectrum of PEDOT-TMe-gCD in DMSO-d6 are lacking in the manuscript or the supporting information.
Answer Q1: We appreciate the validity of this question and we agree with the reviewer’s remark. Now we included in the Supplementary Material the FT-IR spectra (Figure S3) (page 3, bottom) and 1 H NMR spectrum of PEDOT∙TMe-gCD (page 4, top) as advised. We should underline that the ROESY of the inclusion complex of EDOT∙TMe-γCD is not subject of the present manuscript it was discussed in the reference 28.
Q2: I don’t agree with the calculation method for the coverage ratio of PEDOT-TMe-bCD. How many repeating units of PEDOT is sterically covered by a CD? The information should be absolutely included in the manuscript. Figure S3 indicates that the integral ratio between the anomeric protons and the methylene protons of PEDOT is 7 : 11.05. With the integral ratio, they concluded the coverage ratio to be 40%. If the numerical value of coverage ratio is correct, the results mean that one CD must sterically cover approximately 7 repeating unit of PEDOT. For example, it is well-known that an alpha-CD sterically covers two repeating units of PEG in the structure of PR comprising alpha-CD and PEG. Based on the ratio, rotaxane chemists calculate the coverage ratio of PRs as the ratio of sterically covered repeating units to the total repeating units of axle.
Answer Q2: We appreciate the validity of this question and we agree with the reviewer’s remark. Unfortunately, in this time we do not have all the data such as the heights of TMe-bCD and TMe-gCD and PEDOT to have correct information about the numerical value of coverage ratio of PEDOT∙TMe-bCD and PEDOT∙TMe-gCD. It is true that in our previously reported paper (http://dx.doi.org/10.1016/j.eurpolymj.2017.06.015), we calculated correct the numerical value of coverage ratio of PEG (knowing the exact number of ethylene glycol repeating units on the macromolecular chains) having all the data concerning the heights of αCD (7 Å), TMαCD (12 Å) and ethylene glycol units (6.6 Å length). In our case is not possible to estimate the number of EDOT repeat units after polymerization process. We could only estimate the molecular weights from GPC analysis for the soluble fractions in chloroform and the coverage ratio was calculated from 1H NMR spectrum.
The related explanation has not been included in the revised manuscript.
Q3: Considering the vague coverage ratio, I don’t agree with the sentence at line 315: Nevertheless, the 1H NMR spectra of both compounds show broad peaks associated to pai-pai interactions in DMSO-d6 solutions. Because the integral ratio in the 1H NMR spectrum seems to indicate the structure with high coverage ratio, the polymer would have no chance for intermolecular interactions.
Answer Q3: Unfortunately, the presence of non-encapsulated PEDOT chains can give π-π interactions in DMSO-d6 solutions as we and other previously confirmed (see reference 34 and 35 from the Main Ms (https://doi.org/10.1021/acsomega.7b00943 and https://doi.org/10.1002/ejoc.201801724). For this reason we modified the order of references 36 with 35 in the text to support the comments. More than that taking into consideration that PEDOT is disordered material containing polymer chains with various coverage ratios, the encapsulated PEDOT chains can also give rise weak Van der Waals interactions. In any case π-π interactions of PEDOT in solutions are still far from reaching a general consensus.
The related explanation has not been included in the revised manuscript.
Q4: The fluorescent spectra and the CV curves are also lacking, although the results are discussed in the manuscript. The authors should provide the data, otherwise omit the corresponding sentences from the manuscript.
Answer Q4: We agree with the reviewer’s remark and we have now included in red in the Supplementary Material, the florescence spectra in water (Figure S10, bottom, page 6), the florescence spectra in ACN at different excitation wavelengths (Figure S11, top, page 7) and cyclic voltammograms of PEDOT∙TMe-βCD and PEDOT∙TMe-γCD (Figure S12, bottom, page 7). The text introduced in the Main Ms (page 9, bottom) to support the comments is:
FL spectra of PEDOT∙TMe-βCD and PEDOT∙TMe-γCD (λex = 375 nm) exhibited different bands and shoulders in H2O (Figure S10). FL spectra at different excitation wavelengths in ACN (Figure S11) are dependent on the excitation wavelength denoting the existence of a distribution of energetically different molecules in the ground state coupled with a low rate of the excited state relaxation processes.

Round 2
Reviewer 2 Report
I confirmed that the authors have revised most of the points which I indicated. However, a following point should be absolutely revised before the publication. As I previously mentioned, rotaxane chemists use the word "coverage ratio" to indicate the steric coverage of axle by a CD. On the other hand, the authors use the word as the molar ratio between CD and the repeating unit of PEDOT. The word "coverage ratio" is a technical term in the field of rotaxane chemistry. If you want to keep the sentences, you should change the word "coverage ratio" to "molar ratio" thoughout the manuscript.
Author Response
We thank you for the constructive criticism and suggestions, which are valuable and improve the quality of our paper.
Question of Reviewer # 2
Comments and Suggestions for Authors
Q1: I confirmed that the authors have revised most of the points which I indicated. However, a following point should be absolutely revised before the publication. As I previously mentioned, rotaxane chemists use the word "coverage ratio" to indicate the steric coverage of axle by a CD. On the other hand, the authors use the word as the molar ratio between CD and the repeating unit of PEDOT. The word "coverage ratio" is a technical term in the field of rotaxane chemistry. If you want to keep the sentences, you should change the word "coverage ratio" to "molar ratio" thoughout the manuscript.
Answer Q1: We agree with the reviewer’s remark and we changed the word “coverage ratio” in “molar ratio” in the Main Ms in red (page 7, bottom) and the text for the NMR characterization was modified accordingly:
By using the ratio of the integrated area of the H-1 from TMe-βCD (5.05 ppm, IH-1) and the methylene proton peaks of the PEDOT (4.25 ppm, IPEDOT) (IH-1/7)/(IPEDOT/4) the molar ratio in PEDOT∙TMe-βCD was found to be ~ 40.0%. In comparison, the 1H-NMR spectrum of PE-DOT∙TMe-γCD (Figure S5) indicated lower molar ratio ~ 30.0 %.
